# Rapid SARS-CoV-2 Variants Enzymatic Detection (SAVED) by CRISPR-Cas12a

Jun Yang,[a] Nilakshi Barua,[a] Md Nannur Rahman,[a,d] Carmen Li,[a] Norman Lo,[a] Kai Yan Yeong,[a] Tsz Fung Tsang,[a] Xiao Yang,[a] Yuk-Yam Cheung,[b] Alan K. L. Tsang,[b] Rickjason C. W. Chan,[b] Eddie Chi-Man Leung,[c] Paul K. S. Chan,[a] Margaret Ip[a]

[a]Department of Microbiology, Faculty of Medicine, The Chinese University of Hong Kong, Prince of Wales Hospital, Hong Kong, SAR, China
[b]Microbiology Division, Public Health Laboratory Services Branch, Centre for Health Protection, Department of Health, Hong Kong, SAR, China
[c]Department of Microbiology, Prince of Wales Hospital, Hong Kong, SAR, China
[d]Department of Food Technology and Nutritional Science, Mawlana Bhashani Science and Technology University, Tangail, Bangladesh

**ABSTRACT** The continuous and rapid surge of severe acute respiratory syndrome coronavirus 2 (SARS-CoV-2) variants with high transmissibility and evading neutralization is alarming, necessitating expeditious detection of the variants concerned. Here, we report the development of rapid SARS-CoV-2 variants enzymatic detection (SAVED) based on CRISPR-Cas12a targeting of previously crucial variants, including Alpha, Beta, Gamma, Delta, Lambda, Mu, Kappa, and currently circulating variant of concern (VOC) Omicron and its subvariants BA.1, BA.2, BA.3, BA.4, and BA.5. SAVED is inexpensive (US$3.23 per reaction) and instrument-free. SAVED results can be read out by fluorescence reader and tube visualization under UV/blue light, and it is stable for 1 h, enabling high-throughput screening and point-of-care testing. We validated SAVED performance on clinical samples with 100% specificity in all samples and 100% sensitivity for the current pandemic Omicron variant samples having a threshold cycle ($C_T$) value of $\leq 34.9$. We utilized chimeric CRISPR RNA (crRNA) and short crRNA (15-nucleotide [nt] to 17-nt spacer) to achieve single nucleotide polymorphism (SNP) genotyping, which is necessary for variant differentiation and is a challenge to accomplish using CRISPR-Cas12a technology. We propose a scheme that can be used for discriminating variants effortlessly and allows for modifications to incorporate newer upcoming variants as the mutation site of these variants may reappear in future variants.

**IMPORTANCE** Rapid differentiation and detection tests that can directly identify SARS-CoV-2 variants must be developed in order to meet the demands of public health or clinical decisions. This will allow for the prompt treatment or isolation of infected people and the implementation of various quarantine measures for those exposed. We report the development of the rapid SARS-CoV-2 variants enzymatic detection (SAVED) method based on CRISPR-Cas12a that targets previously significant variants like Alpha, Beta, Gamma, Delta, Lambda, Mu, and Kappa as well as the VOC Omicron and its subvariants BA.1, BA.2, BA.3, BA.4, and BA.5 that are currently circulating. SAVED uses no sophisticated instruments and is reasonably priced ($3.23 per reaction). As the mutation location of these variations may reoccur in subsequent variants, we offer a system that can be applied for variant discrimination with ease and allows for adjustments to integrate newer incoming variants.

**KEYWORDS** CRISPR-Cas12a, chimeric crRNA, SARS-CoV-2 variants detection, SAVED, SNP genotyping, short crRNA

Address correspondence to Margaret Ip, margaretip@cuhk.edu.hk.
The authors declare no conflict of interest.

The ongoing pandemic of COVID-19 caused by severe acute respiratory syndrome coronavirus 2 (SARS-CoV-2) has infected more than 559.47 million and killed about 6.36 million people as of 19 July 2022 (1). Vaccines were believed to be able to curb this pandemic. However, the emergence of Alpha (B.1.1.7), Beta (B.1.351), Gamma (P.1),

Delta (B.1.617.2), Omicron (B.1.1.529), Lambda (C.37), and Mu (B.1.621) variants made this impossible through increasing transmissibility and/or resistance to neutralization by antibodies produced from vaccination and even convalescent-phase serum from different strains (2–8). The SARS-CoV-2 spike (S) glycoprotein, especially the receptor-binding domain (RBD), is the main target of neutralizing antibodies (6). However, the emergence of SARS-CoV-2 variants, including a few to over 30 mutations in the spike protein, led to minimal protection for the vaccinated population and the surviving population infected by different variants (5, 9–12). The Omicron variant has become the dominant SARS-CoV-2 variant in the world (13). In addition, the subvariants of Omicron are rapidly evolving. These subvariants have different characteristics and transmissibility, such as Omicron BA.2.12.1, BA.4, and BA.5, which can escape the neutralization of antibodies elicited by Omicron BA.1, repeatedly posing a threat to public health (5, 11, 14). To fulfill the need of public health or clinical decisions, it is imperative to develop rapid differentiation tests that can identify these variants directly, resulting in prompt treatment or isolation of infected individuals and implementation of different quarantine measures for those exposed.

The current identification and tracking of SARS-CoV-2 variants in circulation heavily rely on expensive genomic sequencing and multiple mutation-specific reverse transcription-quantitative PCR (RT-qPCR) as better alternative methods are deficient (15, 16). However, genomic sequencing and RT-qPCR require expensive instrumentation and experienced technicians for result interpretation. This confines variant detection to diagnostic or reference laboratories and compromises the stake of rapid point-of-care (POC) testing. Clustered regularly interspaced short palindromic repeats (CRISPR)-Cas-based platforms are versatile tools for nucleic acid detection, as they are sequence specific, easy to program, and highly efficient. Many CRISPR-Cas diagnostic platforms have been developed for SARS-CoV-2 variant detection, including Cas9 (17), Cas12a (18, 19), and Cas13a (20–22). Compared to Cas12a, Cas9 is more tolerant to one or more mismatches in the spacer, which makes Cas9 difficult for designing CRISPR RNAs (crRNAs) targeting many mutation sites in one go (23, 24). Although Cas13a is more sensitive than Cas12a, it takes a longer incubation time and extra reagents for *in vitro* transcription (IVT). In addition, the reporter of Cas13a is single-stranded RNA (ssRNA), which is prone to false-positive results (21, 25, 26). Thus, Cas12a was selected for this study. Cas12a is sensitive to mismatches, but achieving single-nucleotide polymorphism (SNP) genotyping is still arduous, as mismatch(es) can decrease the signal but cannot abolish it (27, 28). Current studies using CRISPR-Cas12a technology for detection of these variants include the following: (i) detection of Δ144, E484K, and N501Y with only Alpha variant differentiation (19); (ii) a high-fidelity Cas12 enzyme named CasDx1 that was tested on L452R, E484K, and N501Y but cannot be used for variant differentiation due to emergence of the Omicron variant (29); and (iii) two platforms based on reverse transcription PCR (RT-PCR) and CRISPR-Cas12a, which rely on a fluorescence reader to differentiate variants because the wild type and other mutations at the same sites showed false-positive signals, and thus, tube visualization or lateral flow strip are not applicable (18, 30).

Although Cas12a and crRNAs can be engineered to achieve higher specificity, crRNA modification is relatively easier and safer (27–29, 31). Four crRNA designing strategies were employed to improve the specificity of crRNAs in this study, which include (i) chimeric crRNA (replacing the last 8 nucleotide [nt] RNA of the 24-nt spacer with DNA) (28, 32), (ii) short crRNA (15-nt to18-nt spacer) (33), (iii) 3'DNA7 (7-nt DNA 3′ end extension to crRNA, spacer + 3′ end 7 universal DNA extension [TATTATT]) (27), and (iv) introduction of one or two mismatches in the spacer of crRNA. We used LbCas12a in this study, as LbCas12a has stronger collateral cleavage activity (nonspecific single-stranded DNA [ssDNA] degradation) than AsCas12a and FnCas12a (27, 34).

We aimed to provide an affordable assay that required the fewest instruments with a short turnaround time for simultaneous detection of variants. Initially, we focused on improving the specificity of N501Y detection using chimeric crRNA (32). We eventually expanded to include mutations in current SARS-CoV-2 variants. We created the SARS-CoV-2

variants enzymatic detection (SAVED) assay by combining reverse transcription-recombinase polymerase amplification (RT-RPA) and CRISPR-Cas12a to meet the urgent need for a quick, affordable, simple to conduct, and instrument-free detection platform that could detect SARS-CoV-2 and its variants. By employing a plate reader to monitor fluorescence and tube visualization under UV/blue light-emitting diode (LED) light, we confirmed the ability of SAVED to distinguish between Alpha, Beta, Delta, Omicron BA.1, and BA.2 on clinical samples. The turnaround time of the SAVED platform is 70 min, and it can be stable for another 60 min. SAVED showed 100% sensitivity for current pandemic Omicron variant samples having a threshold cycle ($C_T$) of ≤34.9 and 100% specificity for all samples. To our knowledge, SAVED is the first assay that supports both large sample sizes and point-of-care (POC) testing based on CRISPR-Cas12a targeting previously crucial variants, including Alpha, Beta, Gamma, Delta, Lambda, Mu, Kappa, and currently circulating variant of concern (VOC) Omicron and its subvariants BA.1, BA.2, BA.3, BA.4, and BA.5. The scheme that we developed could also be adapted in accordance to current variant proportions (13). We improved the crRNA specificity by chimeric crRNA and short crRNAs (15-nt to 17-nt spacer) for SNP genotyping, which could be applied to the variant differentiation that will be encountered in the future.

## RESULTS

**Combination of mutation sites for variant detection.** We sought to choose unique or important mutation sites to detect the SARS-CoV-2 variants (Fig. 1). To expedite and use fewer reactions for detection, those variants which were considered as VOCs and variants of interest (VOIs) as of 9 March 2022 were divided into the N501Y group (Alpha, Beta, Gamma, Omicron, Mu), T478K group (Delta and Omicron), F490S group (Lambda), and E484A group (Omicron). E484A is unique for Omicron; thus, strains shown to be N501Y, T478K, and E484A positive should be Omicron. The mutation sites Δ144, Δ242–244, D138Y/R190S/K417T, L452R, L452Q/R246N+Δ247–253, and R346K/YY144–145TSN could be used for further differentiation or confirmation of variants of Alpha, Beta, Gamma, Delta, Lambda, and Mu, respectively (Fig. 1). Epsilon, Kappa, and Omicron BA.4 and BA.5 variants also possess L452R. The absence of T478K or presence of E484Q and E484A in these variants facilitated us to differentiate them from the Delta variant (Fig. 1). Zeta possesses only E484K, Eta includes Δ144+E484K, Theta contains N501Y+E484K but lacks Beta and Gamma's specific mutation sites, and Iota possesses T95I but lacks other respective mutation sites (Fig. 1). The K417N and E484K mutations cannot be used for differentiation but are crucial for reducing vaccine neutralization (35).

The Omicron subvariant BA.3 does not show high prevalence for E484A, but it was not prevalent at the time of our assay development (13). The G142D+Δ143–145 (BA.1, BA.1.1, BA.3), G142D (BA.2, BA.2.12.1, BA.4, BA.5), R346K (BA.1.1), L452Q (BA.2.12.1), and L452R (BA.4, BA.5) are the markers of respective subvariants (Fig. 1).

**crRNA optimization for variant detection.** To compare the specificity of crRNA, RPA or RT-RPA was conducted using 1E+09 copies/$\mu$L of synthetic DNA/RNA containing gene fragments of SARS-CoV-2, followed by CRISPR-Cas12a detection. The acceptance of specificity of a crRNA was defined as the fluorescence signal of other mutations or wild-type nucleotide sequences that are lower than 5,000 AU after 1.5 h of incubation at 37°C. Speed acceptance was defined as fluorescence signal over 10,000 AU after 30 min of incubation of the positive samples. We used 5,000 AU as the threshold and validated against spectrophotometry because below 5,000 AU the signal was not visible under the UV light.

We designed a series of crRNAs according to the four strategies mentioned above, targeting selected mutation sites and N2 gene for confirmation (sequences and templates for IVT are shown in Table S1 in the supplemental material). However, some mutation sites were easier to detect with chimeric or short crRNAs and did not require all four strategies for designing the crRNAs. We evaluated the performance of these crRNAs (see Fig. S1 to S20 in the supplemental material) and selected a set of crRNAs based on specificity and reaction speed (Fig. 2). We successfully optimized all crRNAs of selected mutation sites using chimeric or short (15-nt to 17-nt spacer) crRNAs for SNP genotyping and

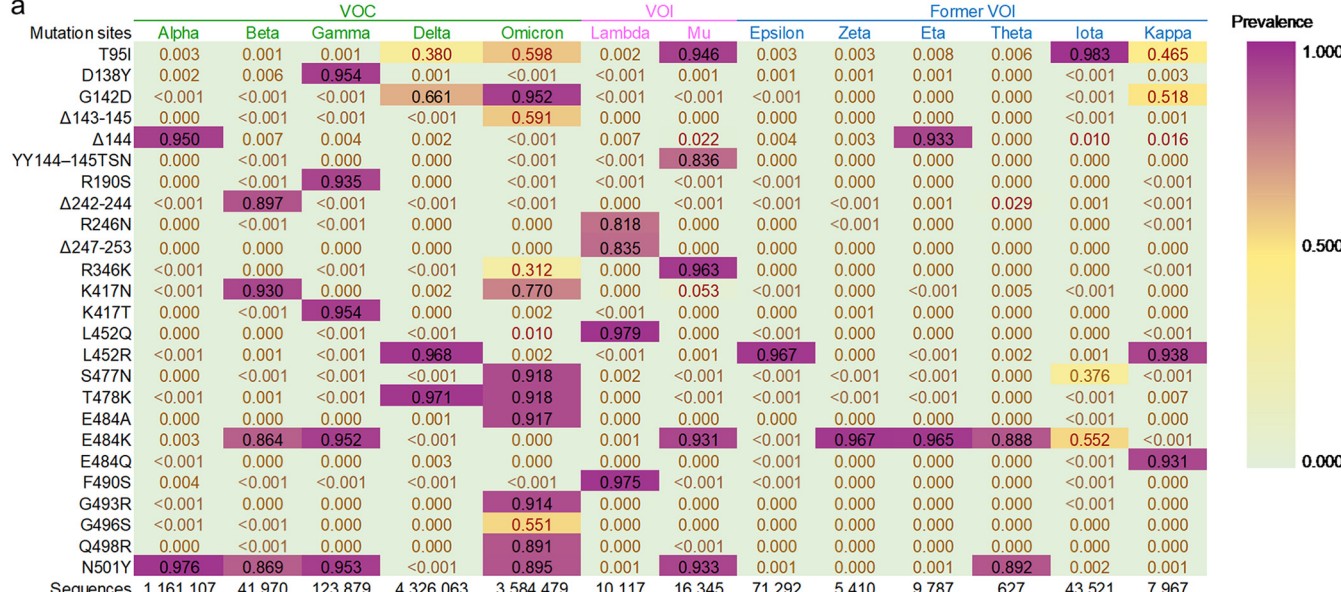

**FIG 1** Mutation sites comparison for variant detection and differentiation. (a) Comparison of mutation sites of crucial variants. (b) Comparison of mutation sites of important Omicron subvariants. The worldwide prevalence of mutation sites is represented by the color change from pale green to purple. The data for comparing the mutation sites in the SARS-CoV-2 variants were extracted from the outbreak info (44) and WHO (45).

regular crRNAs for insertion/deletions. The types and sequences of some selected crRNAs aligned to corresponding mutation sites, and wild-type sequences are shown in Fig. 3.

Among these selected crRNAs, D138Y crRNA (12th) 15 nt and G142D crRNA (13th) 15 nt showed better specificity when using 40 nM Cas12a compared with 100 nM (Fig. S2c, S2d*, S3c, and S3d*). Chimeric crRNA 24-nt and 16-nt spacer crRNA can differentiate S477N+T478K from T478K, Q498R+N501Y from N501Y, respectively, using RT-RPA to amplify the RNA but not DNA samples before CRISPR-Cas12a detection (Fig. S14a, S14e, S20e, and S20g). Thus, the Omicron variant was excluded from the N501Y and T478K groups.

**Limit of detection determination and specificity of CRISPR-Cas12a-based detection.** We determined limit of detection (LoD) with RT-RPA products using serial dilutions of IVT RNA containing gene fragments of SARS-CoV-2 variants (Table 1). A total of 1E+09

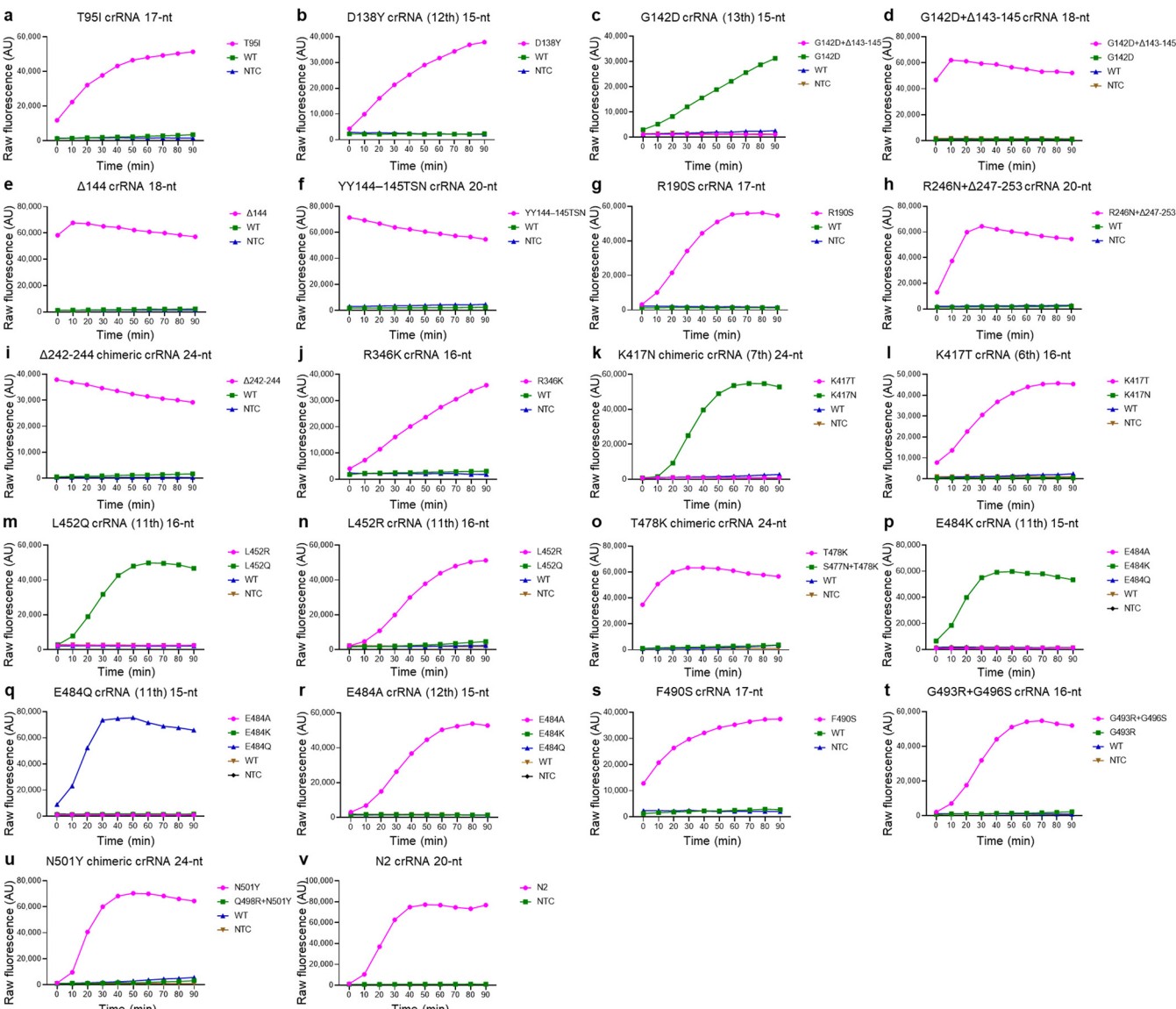

**FIG 2** Selected crRNAs for SARS-CoV-2 variant detection. crRNAs fluorescence signal during 1.5 h of incubation with 100 nM Cas12a and 200 nM crRNA, except panels b, c, h, i, k, l, and v with 40 nM Cas12a and 40 nM crRNA. (a) T95I crRNA 17 nt. (b) D138Y crRNA (12th) 15 nt. (c) G142D crRNA (13th) 15 nt. (d) G142D+Δ143-145 crRNA 18 nt. (e) Δ144 crRNA 18 nt. (f) YY144–145TSN crRNA 20 nt. (g) R190S crRNA 17 nt. (h) R246N + Δ247–253 crRNA 20 nt. (i) Δ242–244 chimeric crRNA 24 nt. (j) R346K crRNA 16 nt. (k) K417N chimeric crRNA (7th) 24 nt. (l) K417T crRNA (6th) 16 nt. (m) L452Q crRNA (11th) 16 nt. (n) L452R crRNA (11th) 16 nt. (o) T478K chimeric crRNA 24 nt. (p) E484K crRNA (11th) 15 nt. (q) E484Q crRNA (11th) 15 nt. (r) E484A crRNA (12th) 15 nt. (s) F490S crRNA 17 nt. (t) G493R+G496S crRNA 16 nt. (u) N501Y chimeric crRNA 24 nt. (v) N2 crRNA 20 nt. RPA using 1E+09 copies/μL of synthetic DNA containing gene fragments of SARS-CoV-2 for CRISPR-Cas12a detection, except panels o and u where RT-RPA was used with 1E+09 copies/μL RNA of S477N+T478K and Q498R+N501Y. AU, arbitrary units.

copies/μL RNA of wild type or different mutations at the same sites was used as a template to evaluate the specificity of these selected crRNAs. We verified the reproducibility of the LoD of all CRISPR-Cas12a-based mutation detection under both fluorescence readouts and UV light at 37°C for 1.5 h of incubation. The LoD values of selected reactions were 1E+01 copies/μL and even further lower to 1E+0 copies/μL for Δ242–244 chimeric crRNA 24 nt and R246N+Δ247–253 crRNA 20 nt (Table 1; see also Fig. S21 in the supplemental material). Results of incubation from 30 min to 1.5 h were the same, so 30 min was selected as the minimum incubation time for further detection. We demonstrated that CRISPR-Cas12a-based mutation detection of all selected crRNAs was specific to corresponding mutations with no cross activity for other mutations at the same sites and wild types (Table 1; Fig. S21). We did not

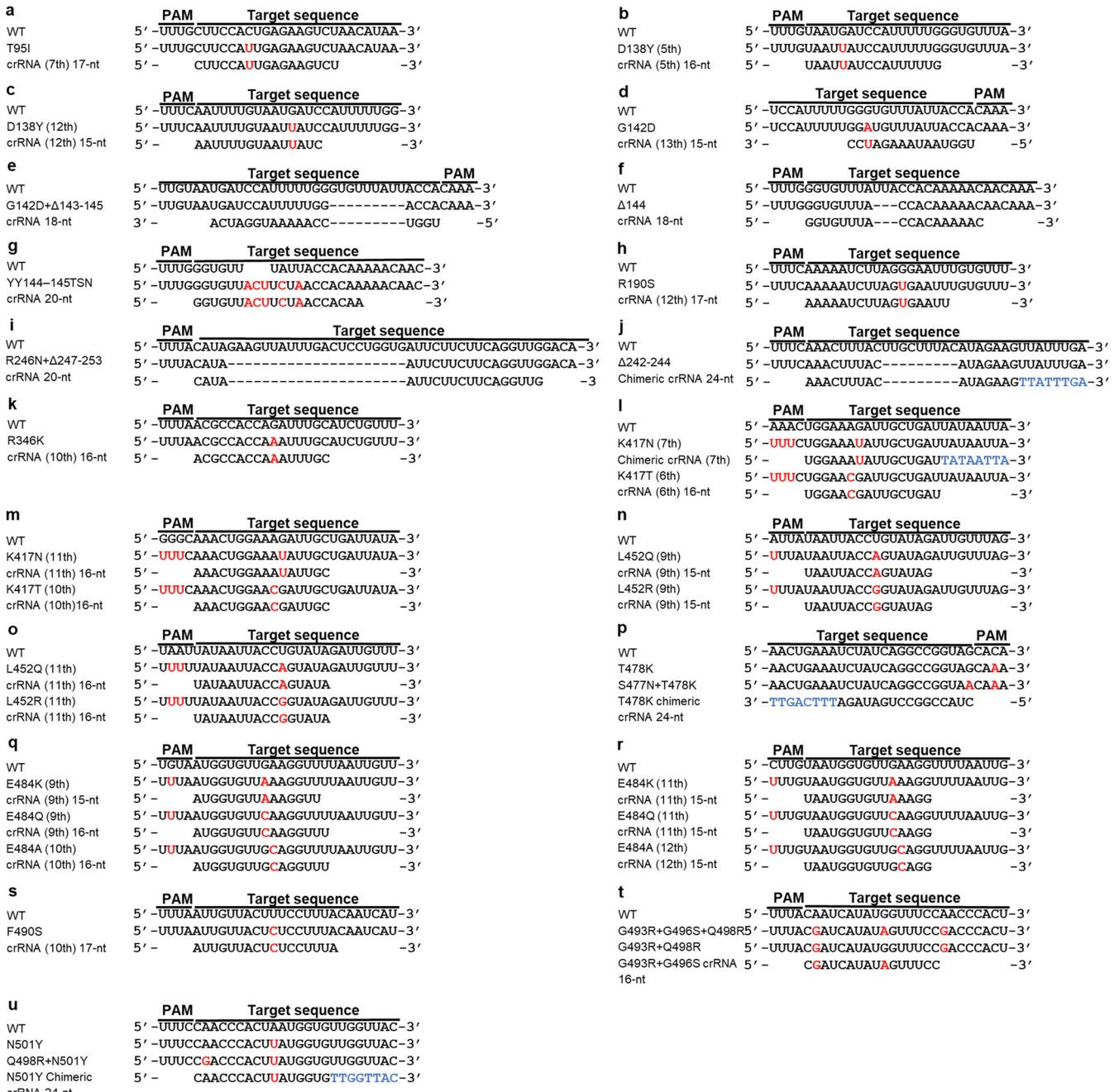

**FIG 3** Alignment of crRNAs to respective mutation sites and wild-type sequences. (a) T95I crRNA (7th) 17 nt. (b) D138Y crRNA (5th) 16 nt. (c) D138Y crRNA (12th) 15 nt. (d) G142D crRNA (13th) 15 nt. (e) G142D+Δ143-145 crRNA 18 nt. (f) Δ144 crRNA 18 nt. (g) YY144–145TSN crRNA 20 nt. (h) R190S crRNA (12th) 17 nt. (i) R246N+Δ247–253 crRNA 20 nt. (j) Δ242–244 chimeric crRNA 24 nt. (k) R346K crRNA (10th) 16 nt. (l) K417N chimeric crRNA (7th) 24 nt and K417T crRNA (6th) 16 nt. (m) K417N crRNA (11th) 16 nt and K417T crRNA (10th) 16 nt. (n) L452Q crRNA (9th) 15 nt and L452R crRNA (9th) 15 nt. (o) L452Q crRNA (11th) 16 nt and L452R crRNA (11th) 16 nt. (p) T478K chimeric crRNA 24 nt. (q) E484K crRNA (9th) 15 nt, E484Q crRNA (9th) 16 nt, and E484A crRNA (10th) 16 nt. (r) E484K crRNA (11th) 15 nt, E484Q crRNA (11th) 15 nt, and E484A crRNA (12th). (s) F490S crRNA 17 nt. (t) G493R+G496S crRNA 16 nt. (u) N501Y chimeric crRNA 24 nt. The red font shows the mutation sites and introduced protospacer-adjacent motifs (PAMs), whereas the blue font shows RNA nucleotides replaced by DNA.

conduct LoD determination on T95I, R190S, and R346K, as none of the primer sets showed amplification at 1E+03 copies/μL.

**Scheme of variants detection.** Extracted RNA from clinical samples was used for RT-RPA using N501Y, T478K, and E484A RT-RPA primers. For RT-RPA of F490S, the same primers as N501Y were used. RT-RPA products were detected separately by their respective crRNAs (Fig. 4 and Table 1). N501Y-positive strains were tested by Δ144 crRNA

**TABLE 1** LoD and specificity of selected SAVED CRISPR-Cas12a-based fluorescence detections after incubation at 37°C for 30 min[a]

| Corresponding mutation(s) | RNA (copies/$\mu$L) | | | | | | 1E+09 (WT/other mutations) | NA (NTC) |
|---|---|---|---|---|---|---|---|---|
| | 1E+05 | 1E+04 | 1E+03 | 1E+02 | 1E+01 | 1E+00 | | |
| D138Y crRNA (12th) 15 nt | 9/9 | 9/9 | 9/9 | 9/9 | **9/9** | 0/9 | 0/9 | 0/9 |
| G142D crRNA (13th) 15 nt | 9/9 | 9/9 | 9/9 | 9/9 | **9/9** | 3/9 | 0/9 | 0/9 |
| G142D+Δ143–145 crRNA 18 nt | 9/9 | 9/9 | 9/9 | 9/9 | **9/9** | 0/9 | 0/9 | 0/9 |
| Δ144 crRNA 18 nt | 9/9 | 9/9 | 9/9 | 9/9 | **9/9** | 3/9 | 0/9 | 0/9 |
| YY144–145TSN crRNA 20 nt | 9/9 | 9/9 | 9/9 | 9/9 | **9/9** | 3/9 | 0/9 | 0/9 |
| Δ242–244 chimeric crRNA 24 nt | 9/9 | 9/9 | 9/9 | 9/9 | 9/9 | **9/9** | 0/9 | 0/9 |
| R246N+Δ247–253 crRNA 20 nt | 9/9 | 9/9 | 9/9 | 9/9 | 9/9 | **9/9** | 0/9 | 0/9 |
| K417N chimeric crRNA (7th) 24 nt | 9/9 | 9/9 | 9/9 | 9/9 | **9/9** | 0/9 | 0/9 | 0/9 |
| K417T crRNA (6th) 16 nt | 9/9 | 9/9 | 9/9 | 9/9 | **9/9** | 0/9 | 0/9 | 0/9 |
| L452Q crRNA (11th) 16 nt | 9/9 | 9/9 | 9/9 | 9/9 | **9/9** | 0/9 | 0/9 | 0/9 |
| L452R crRNA (11th) 16 nt | 9/9 | 9/9 | 9/9 | 9/9 | **9/9** | 0/9 | 0/9 | 0/9 |
| T478K chimeric crRNA 24 nt | 9/9 | 9/9 | 9/9 | 9/9 | **9/9** | 0/9 | 0/9 | 0/9 |
| E484A crRNA (12th) 15 nt | 9/9 | 9/9 | 9/9 | 9/9 | **9/9** | 6/9 | 0/9 | 0/9 |
| E484K crRNA (11th) 15 nt | 9/9 | 9/9 | 9/9 | 9/9 | **9/9** | 0/9 | 0/9 | 0/9 |
| E484Q crRNA (11th) 15 nt | 9/9 | 9/9 | 9/9 | 9/9 | **9/9** | 0/9 | 0/9 | 0/9 |
| F490S crRNA 17 nt | 9/9 | 9/9 | 9/9 | 9/9 | **9/9** | 0/9 | 0/9 | 0/9 |
| G493R+G496S crRNA 16 nt | 9/9 | 9/9 | 9/9 | 9/9 | **9/9** | 3/9 | 0/9 | 0/9 |
| N501Y chimeric crRNA 24 nt | 9/9 | 9/9 | 9/9 | 9/9 | **9/9** | 0/9 | 0/9 | 0/9 |
| N2 crRNA 20 nt | 9/9 | 9/9 | 9/9 | 9/9 | **9/9** | 0/9 | NA | 0/9 |

[a]The CRISPR-Cas12a mixture contained 100 nM Cas12a and 200 nM crRNA except D138Y, G142D, Δ242–244, and R246N+Δ247–253 with 40 nM Cas12a and 40 nM crRNA. NTC stands for nontemplate control. NA indicates not applicable. We conducted measurements in triplicates, each with three technical replicates ($n = 9$); tube results under UV light are not included. The lowest concentrations with 100% sensitivity are shown in bold font.

18 nt, Δ242–244 chimeric crRNA 24 nt, D138Y crRNA (12th) 15 nt, and YY144–145TSN crRNA 20 nt for differentiation to Alpha, Beta, Gamma, and Mu variants, respectively. As the Theta variant has become rare, we did not include it in the scheme (Fig. 4). T478K-positive strains were confirmed by L452R crRNA (11th) 16 nt. The K417N mutation can be tested to identify Delta plus; however, we did not include it because of its

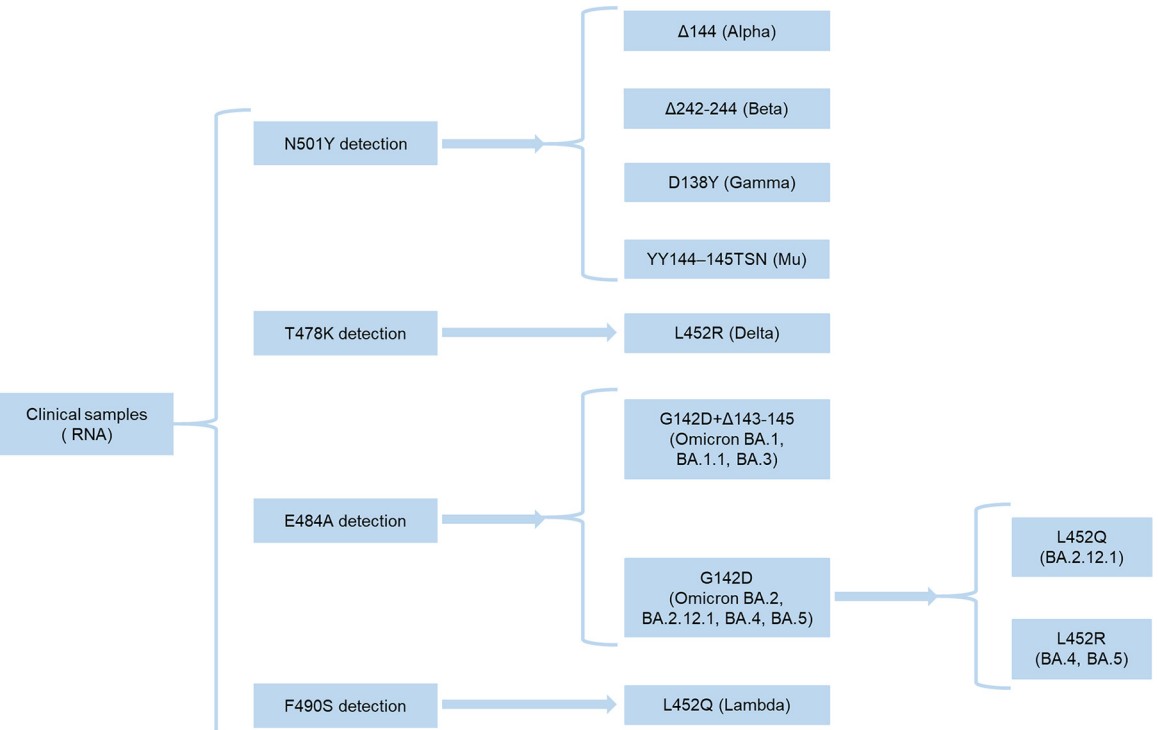

**FIG 4** Scheme to illustrate the envisioned detection pipeline of SARS-CoV-2 variant detection.

scarcity. Detection of E484A indicates the Omicron variant, which was then divided into two parts. Part 1 included BA.1, BA.1.1, and BA.3, which was detected by G142D+Δ143–145 crRNA 18 nt. Part 2 had BA.2, BA.2.12.1, BA.4, and BA.5 when G142D positive, which was further divided into BA.2.12.1 (L452Q positive), BA.4, and BA.5 (L452R positive). Although the Omicron variant possesses the N501Y and T478K mutation sites, our scheme uses E484A detection to confirm Omicron as the mismatches in the RT-RPA primers, and crRNAs of N501Y and T478K give signals less than the threshold for detection. The presence of F490S proves the presence of Lambda, which could be confirmed by using L452Q crRNA (11th) 16 nt. As the Omicron variant is dominant globally, we can shorten this scheme to Omicron subvariants only (13).

**Validation of SAVED CRISPR-Cas12a-based detection using clinical samples.** To evaluate the performance of SAVED CRISPR-Cas12a-based detection on clinical samples, the samples were randomized and blinded to research staff to minimize detection bias (details in Table S2 in the supplemental material). As we did not have Gamma, Mu, and Lambda variants, our first screening only included N501Y, T478K, and E484A detections; the second screening contained Δ144, Δ242–244, L452R, G142D+Δ143–145, and G142D (Fig. 5). In parallel, we conducted SAVED detection on clinical samples by fluorescence and tube visualization by UV light, which showed the same result. The detection results were compared with $C_T$ values of RT-qPCR for LoD determination in clinical samples. We identified all negative samples for each mutation site using our method, and the specificity was 100% for all detection that we tested (Fig. 5 and Table 2; see also Fig. S22 to S24 in the supplemental material). All Alpha and Beta samples with $C_T$ values of < 32.8 (≤30.3; no samples 30.3 to 32.8) were detectable by N501Y detection (Fig. 5 and Table 2; Fig. S22). All Delta samples with $C_T$ values of <31.1 (≤ 30.8) were identified by T478K and L452R detection (Fig. 5 and Table 2; Fig. S23). All Omicron variants can be confirmed by E484A detection (Fig. 5 and Table 2; Fig. S24). The Δ144, Δ242–244, and G142D+Δ143–145 exhibited 100% sensitivity, whereas one sample of G142D detection showed false-negative signal ($C_T$, 33.2) (Fig. 5 and Table 2; Fig. S22 to S24). We also conducted SARS-CoV-2 confirmation using N2 gene detection, and we identified all samples at $C_T$ values of ≤30.2 (Table 2; see also Fig. S25 in the supplemental material).

## DISCUSSION

We report for the first time that we have achieved rapid detection by fluorescence and tube visualization under UV/blue LED light of all crucial SARS-CoV-2 variants to date using RT-RPA followed by CRISPR-Cas12a. SAVED costs US$3.23 per reaction, whereas RT-qPCR costs US$2.74 per reaction (see Table S3 in the supplemental material). Results can be read within 70 min and stable for 1 h. The CRISPR-Cas12a-based detection period of our platform is 30 min, though we used 1.5 h of incubation to check whether false-positive signals will appear in the wild types or other mutations at the same sites. These ensure signal stability and accuracy when a large sample size needs to be detected. We defined specificity as the absence of a false-positive signal instead of a fold change because a high fold change does not ensure the absence of a false-positive signal. Our scheme used fewer reactions for fast detection and can be modified or shortened for new variants according to the current variant proportions (Fig. 4). For example, according to COVID Data Tracker, CDC (accessed on 18 June 2022) (13), there were only the Omicron BA.2, BA.2.12.1, BA.4, and BA.5 variants. Thus, we can only work on detecting G142D, L452Q, and L452R, resulting in easy differentiation. We evaluated our platform on clinical samples using fluorescence and tube visualization (Fig. 5; see also Fig. S22 to S25 in the supplemental material). Our platform achieved 100% specificity for all clinical samples, 100% sensitivity for all Omicron variant samples, and 100% sensitivity for all detections when samples had a $C_T$ value of ≤30.2. The sensitivity was not high compared to COVID-19 detection only because variants detecting RT-RPA primers must surround the mutation sites as the length limit of RT-RPA (80 to 140 bp) (26, 36). Fluorescence detection is suitable for high-throughput

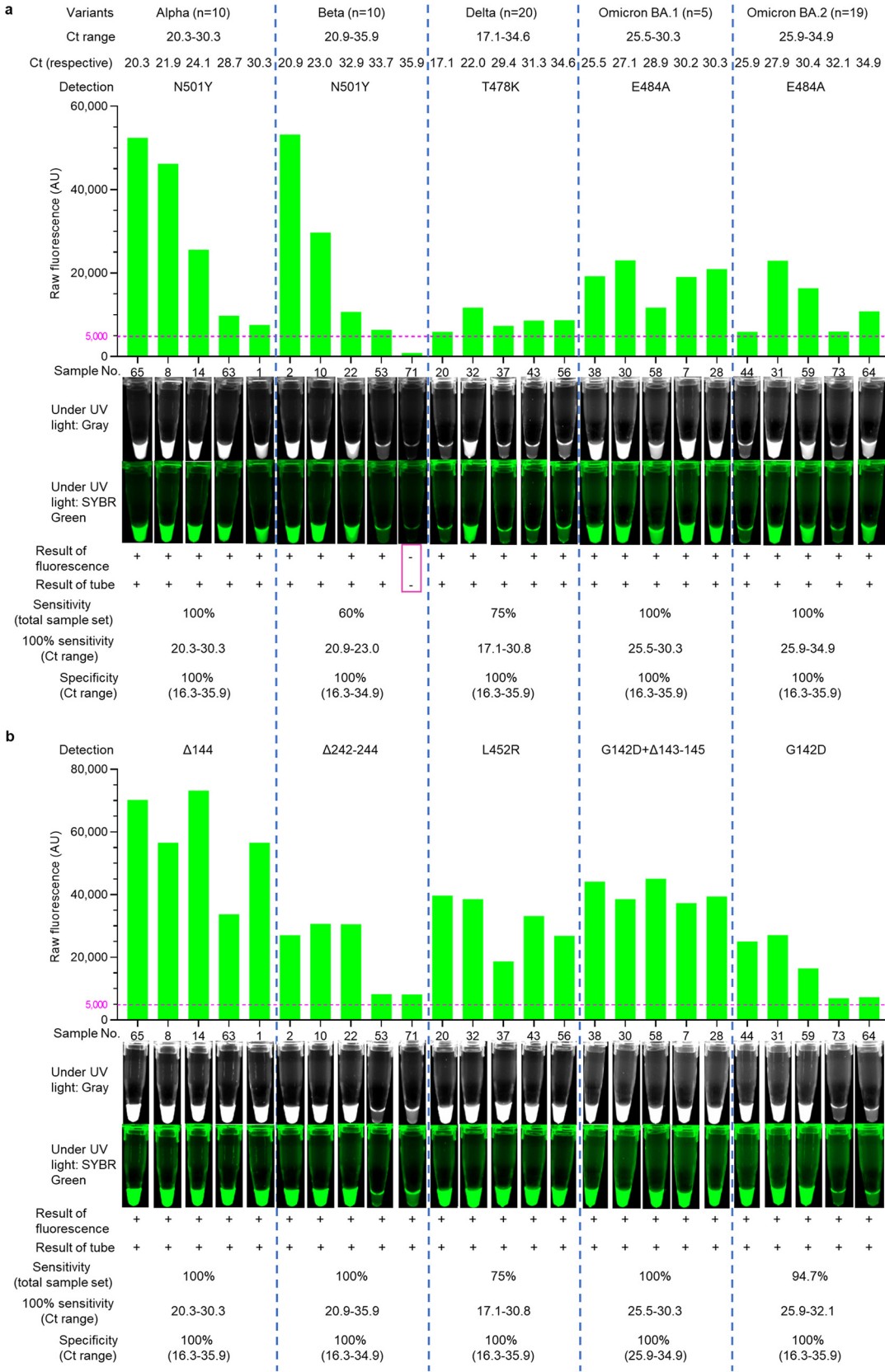

**FIG 5** SAVED CRISPR-Cas12a-based detection in blinded 79 samples, including 64 SARS-CoV-2 variants, 5 wild types, and 10 NTC. Five different SARS-CoV-2 variants, wild type, and NTC were used for clinical sample validation of SAVED CRISPR-Cas12a-based detection.

sample sizes. Blue LED light or UV light can also be used to visualize detection tubes, which enables instrument-free detection (32, 37). We used tube visualization to read out results instead of lateral flow strips, as the tube visualization apparatus (blue LED light source and transparent orange acrylic sheet) is generally inexpensive and can be recycled (32, 37). This is especially important in resource-poor settings.

Since we worked to create a platform that can be read through tube visualization, in addition to specificity, the reaction speed should be as quick as possible. According to our result on SNP genotyping, we found that the fluorescence signal decreased while spacer length decreased (24 nt ≈ 20 nt ≥ 18 nt > 17 nt > 16 nt > 15 nt). Thus, it is necessary to use 17 nt to 19 nt of the spacer to ensure robust activity of Cas12a (23). However, specificity increased while spacer length decreased (see Fig. S1 to S20 in the supplemental material). We found chimeric crRNAs showed stronger signal but weaker specificity than 16-nt spacer crRNAs (Fig. S10 to S16 and S20). According to our results, we can only achieve SNP genotyping by chimeric crRNAs and 15-nt to 17-nt spacer crRNAs, which indicates that the specificity of crRNAs with 18-nt spacers is lower than chimeric crRNAs. We also found the specificity and reaction speed of chimeric and 17-nt spacer crRNAs were comparable because chimeric crRNA did not always show higher specificity and lower reaction speed (Fig. S10a, S10d, S11b, S11e, S11g, S11i, S14a, S14d, S20e, S20f). Only K417T, L452R, and L452Q used the 3'DNA7 strategy, which could amplify signals as previously reported (27). However, it could not improve specificity compared to that of chimeric crRNAs and short crRNAs (15-nt or 16-nt spacer) for these three mutation sites (Fig. S11k, S12d, S12e, S12k, S12l, S13b, S13c, S13g, S13l to o). The introduction of mismatches in regular-length crRNAs could marginally improve specificity but was not adequate for our criteria. Therefore, we tried to introduce mismatch(es) in the 16-nt spacer, and the detection signal was almost abolished (Fig. S11m, S11n, S12e, S12f, S13h to j, S13o to v, S13x to ab).

As previously reported, LbCas12a is sensitive to single mismatches in positions 1 to 17 of the spacer, and two consecutive mismatches in positions 1 to 18 will result in complete loss of activity; however, positions 18 to 23 are highly tolerant to mismatch (es) (23, 24), and substitutions at positions 6 to 11 are more sensitive than other sites (28, 38). However, some studies showed SNP in the protospacer-adjacent motif (PAM) or positions 1 to 7 will lead to a clearly decreased fluorescence signal for the full-length spacer (24 nt). Whereas, the influence of SNP position varied greatly for short crRNAs (16-nt to 17-nt spacer) (33, 36). We tested the performance of short crRNAs (16-nt or 15-nt spacer) at positions 1 to 4, T478K and S477N+T478K (position 1), N501Y and Q498R+N501Y (position 2), and G142D (position 4), and none was acceptable in terms of specificity. However, for D138Y (position 5), the 16-nt spacer crRNA had good specificity but a low reaction speed (Fig. 3; see also Fig. S2a, S3a, S3b, S14, S20e, S20g). For SNPs in position 6, we tested on K417T, E484K, and E484Q, and only K417T crRNA 16 nt was acceptable in terms of specificity, but for E484K and E484Q, neither was acceptable, even the 16-nt spacer. This may indicate that position 6's specificity is not high. Many positions were available for introducing PAMs in RT-RPA primers, and we observed that the specificity varied greatly based on the mutation position sites as previously reported (33, 36). For example, the specificity changed with the positions accordingly in K417N (7 > 11), K417T (6 < 10), L452Q and L452R (11 > 10 > 9 > 7 > 12), E484K (6 < 9 < 11), and E484Q (6 < 11 < 9) (Fig. S10 to S13, S15, and S16). We also observed SNP-introduced PAM in E484K (−2nd), no crRNA, even the 16-nt spacer, exhibited acceptable specificity, which is different from previous study (19),

**FIG 5** Legend (Continued)
We selected 5 representative clinical samples from each variant. Respective $C_T$ values of each sample is mentioned in the figure. Fluorescence detection and tube visualization results of representative samples are shown in the figure. The complete data sets are shown in Figs. S22 to S24. (a) N501Y, T478K, and E484A detections of representative clinical samples. (b) Δ144 (Alpha), Δ242–244 (Beta), L452R (Delta), G142D+Δ143–145 (Omicron BA.1, BA.1.1, and BA.3), and G142D (Omicron BA.2, BA.2.12.1, BA.4, and BA.5) detections after the first-step screening. AU, arbitrary units. The horizontal pink dash line indicates the positive samples' fluorescence threshold (5,000 AU), which can be visualized under UV light. The pink rectangle highlights the false-negative sample.

**TABLE 2** Comparison between SAVED CRISPR-Cas12a detection and $C_T$ value[a]

| Mutation(s) | Detection | All samples | | Samples within specific $C_T$ range (positive) |
|---|---|---|---|---|
| | | Positive | Negative | |
| N501Y | No. positive | 16 | 0 | 12 |
| | No. negative | 4 | 59 | 0 |
| | Total no. | 20 | 59 | 12 |
| | Sensitivity (%) (95% CI) | 80.00 (56.34–94.27) | NA | 100.00 (73.54–100.00) |
| | Specificity (%) (95% CI) | NA | 100.00 (93.94–100.00) | NA |
| | $C_T$ range | 20.3–35.9 | 16.3–34.9 | 20.3–30.3 |
| T478K | No. positive | 15 | 0 | 13 |
| | No. negative | 5 | 59 | 0 |
| | Total no. | 20 | 59 | 13 |
| | Sensitivity (%) (95% CI) | 75.00 (50.90–91.34) | NA | 100.00 (75.30–100.00) |
| | Specificity (%) (95% CI) | NA | 100.00 (93.94–100.00) | NA |
| | $C_T$ range | 17.1–34.6 | 16.3–35.9 | 17.1–30.8 |
| E484A | No. positive | 24 | 0 | 24 |
| | No. negative | 0 | 55 | 0 |
| | Total no. | 24 | 55 | 24 |
| | Sensitivity (%) (95% CI) | 100.00 (85.75–100.00) | NA | 100.00 (85.75–100.00) |
| | Specificity (%) (95% CI) | NA | 100.00 (93.51–100.00) | NA |
| | $C_T$ range | 25.5–34.9 | 16.3–35.9 | 25.5–34.9 |
| Δ144 | No. positive | 10 | 0 | 10 |
| | No. negative | 0 | 10 | 0 |
| | Total no. | 10 | 10 | 10 |
| | Sensitivity (%) (95% CI) | 100.00 (69.15–100.00) | NA | 100.00 (69.15–100.00) |
| | Specificity (%) (95% CI) | NA | 100.00 (69.15–100.00) | NA |
| | $C_T$ range | 20.3–30.3 | 20.9–35.9 | 20.3–30.3 |
| Δ242–244 | No. positive | 10 | 0 | 10 |
| | No. negative | 0 | 10 | 0 |
| | Total no. | 10 | 10 | 10 |
| | Sensitivity (%) (95% CI) | 100.00 (69.15–100.00) | NA | 100.00 (69.15–100.00) |
| | Specificity (%) (95% CI) | NA | 100.00 (69.15–100.00) | NA |
| | $C_T$ range | 20.9–35.9 | 20.3–30.3 | 20.9–35.9 |
| L452R | No. positive | 15 | NA | 13 |
| | No. negative | 5 | NA | 0 |
| | Total no. | 20 | NA | 13 |
| | Sensitivity (%) (95% CI) | 75.00 (50.90–91.34) | NA | 100.00 (75.30–100.00) |
| | Specificity (%) (95% CI) | NA | NA | NA |
| | $C_T$ range | 17.1–34.6 | NA | 17.1–30.8 |
| G142D+Δ143−145 | No. positive | 5 | 0 | 5 |
| | No. negative | 0 | 19 | 0 |
| | Total no. | 5 | 19 | 5 |
| | Sensitivity (%) (95% CI) | 100.00 (47.82–100.00) | NA | 100.00 (47.82–100.00) |
| | Specificity (%) (95% CI) | NA | 100.00 (82.35–100.00) | NA |
| | $C_T$ range | 25.5–30.3 | 25.9–34.9 | 25.5–30.3 |
| G142D | No. positive | 18 | 0 | 16 |
| | No. negative | 1 | 5 | 0 |
| | Total no. | 19 | 5 | 16 |
| | Sensitivity (%) (95% CI) | 94.74 (73.97–99.87) | NA | 100.00 (79.41–100.00) |
| | Specificity (%) (95% CI) | NA | 100.00 (47.82–100.00) | NA |
| | $C_T$ range | 25.9–34.9 | 25.5–30.3 | 25.9–32.1 |
| N2 | No. positive | 48 | 0 | 38 |
| | No. negative | 21 | 10 | 0 |
| | Total no. | 69 | 10 | 38 |
| | Sensitivity (%) (95% CI) | 69.57 (57.31–80.08) | NA | 100.00 (90.75–100.00) |
| | Specificity (%) (95% CI) | NA | 100.00 (69.15–100.00) | NA |
| | $C_T$ range | 16.3–35.9 | NA | 16.3–30.2 |

[a]In total, 79 blinded samples (10 Alpha, 10 Beta, 20 Delta, 5 Omicron BA.1, 19 Omicron BA.2, 5 wild-type strains, and 10 NTC) were detected by a fluorescence reader in a 384-well plate and visualized under UV light in 0.2-mL 8-strip tubes. As fluorescence detection and UV visualization results are the same for incubation from 30 min to 1.5 h, this table only shows the results after 30 min incubation without mentioning which detection method was used. NA, not applicable; CI, confidence interval (26).

due to longer incubation time for crRNA optimization, which allows the appearance of false-positive signal.

From the mutation sites included in this study, we established the specificity of our crRNAs for SNP in positions 7 to 13 using chimeric or short crRNAs. HOLMES v1 suggested using 17-nt spacer crRNA for first detection and then 16 nt and 15 nt (33). We suggest to first use the 16-nt spacer crRNA targeting different SNP positions (7 to 13) to choose the position with the highest specificity and then decide whether it is necessary to increase specificity using 15-nt spacer crRNA or amplify the signal by using either the 17-nt spacer or chimeric crRNA.

We observed that RT-RPA is relatively sensitive to primer mismatches, and only chimeric crRNA 24-nt and 16-nt spacer crRNA can differentiate S477N+T478K from T478K and Q498R+N501Y from N501Y using RNA samples. This could be attributed to the cumulative effect of the mismatches in the primers and crRNAs causing significant reductions in the fluorescence signals (Fig. S14a, S14e, S20e, S20g). So, we used degenerate primers to achieve higher sensitivity for primers with introduced PAMs. T478K RPA R has three mismatches for Omicron BA.1 and BA.1.1 (positions 8, 16, and 23 from the 3′ end); two mismatches for Omicron BA.2 and BA.3 (positions 8 and 23 from the 3′ end); and one mismatch for Omicron BA.4 and BA.5 (positions 23 from 3′ end). When we conducted clinical validation, we did not have BA.4 and BA.5 variants to verify whether T478K detection should be positive or not. The E484A detection could still be used for differentiation even though T478K detection might be positive.

The limitation of variant detection is that we cannot predict and detect new variants when we do not know the sequence. The scheme of variant detection may need to be changed if new variants appear. Moreover, some mutation sites cannot meet the high prevalence or unique requirement, such as YY144–145TSN ($<$85%) and T95I.

To our knowledge, no established methods have achieved the differentiation of all current SARS-CoV-2 variants by using CRISPR-Cas12a due to its specificity limitation. Our platform ensures the differentiation of all current SARS-CoV-2 variants with high specificity and sensitivity through modification of crRNA, and it is rapid, stable, inexpensive, easy-to-use, and instrument free. Moreover, the reaction temperature is 37°C, so it can be employed in low-resource areas where body heat can be used for the reaction (21). The crRNA optimization strategy can be used for other pathogen variant detection, especially for SNP genotyping. Our platform can be further developed for POC diagnostics and high-throughput screening using established platforms (19, 22). CRISPR-Cas12a detection relies on Cas12a's collateral activity once CRISPR-Cas12a achieves target recognition; thus, the strategy of improving crRNA's specificity can also be used for genome editing and gene therapy with single-base precision by CRISPR-Cas12a (27, 28, 39).

## MATERIALS AND METHODS

**Variants, RT-RPA primers, and crRNAs.** SARS-CoV-2 wild-type (GenBank accession number NC_045512.2) (40) and variant sequences were downloaded from NCBI GenBank and GISAID (https://gisaid.org/). The SARS-CoV-2 sequences were aligned by Geneious Prime 2021 to identify the conserved sequences. The spacers of crRNAs were designed and selected following inherent or introduced 5'TTTV PAM. The NCBI Primer-BLAST website was used for RPA primer design according to the TwistAmp assay design manual (41). Information on RT-RPA primers, crRNA IVT templates, and crRNAs of this study are provided in Table S1 and Table S4 in the supplemental material. To verify SARS-CoV-2 samples, we designed RT-RPA primers compatible with a previously reported crRNA of the N2 region of N gene (42).

**Target RNA sample preparation, clinical sample collection, ethics statement, and crRNA preparation.** Target RNA samples were prepared by *in vitro* transcription (IVT) (HiScribe T7 quick high yield RNA synthesis kit; NEB) using purified PCR products with T7 promoter amplified from synthetic gene fragments of SARS-CoV-2 (see Table S5 in the supplemental material) according to the manufacturer's protocol with a few modifications. The PCR was conducted using iProof high-fidelity PCR kit (Bio-Rad) containing 1× iProof HF buffer, 0.2 mM deoxynucleoside triphosphate (dNTP) mix (10 mM each; NEB), 0.5 $\mu$M forward primers (with T7 promoter) and reverse primers, 0.02 U/$\mu$L iProof high-fidelity DNA polymerase, and 1 ng synthetic gene fragments of SARS-CoV-2. The thermal cycling condition was 30 s at 98°C for initial denaturation; 30 cycles of 10 s at 98°C for denaturation, 30 s at 50°C for annealing, and 30 s at 72°C for extension; and 10 min at 72°C for final extension. The PCR products were confirmed by agarose gel electrophoresis and purified by QIAquick gel extraction kit (Qiagen). The IVT reaction mixture containing 10 $\mu$L rNTP buffer mix (10 mM each NTP final), 8 $\mu$L purified PCR products mentioned

above, and 2 $\mu$L T7 RNA polymerase mix was incubated overnight at 37°C. The DNA template in the IVT products was digested by TURBO DNA-free kit (Thermo Fisher). In brief, 20 $\mu$L IVT products was incubated with 24 $\mu$L $H_2O$, 5 $\mu$L 10× TURBO DNase buffer, and 1 $\mu$L TURBO DNase at 37°C for 30 min. Five microliters of resuspended DNase inactivation reagent was then added and incubated for 5 min with three times flicking at room temperature. The samples were then centrifuged at 10,000 × $g$ for 1.5 min at room temperature, and the supernatant containing the RNA was then purified by Monarch RNA cleanup kit (NEB) following the manufacturer's protocol. The concentration and quality of the RNA were measured by NanoDrop 2000 spectrophotometer (Thermo Fisher) and agarose gel electrophoresis. The copy number was calculated by NEBioCalculator (43).

SARS-CoV-2 clinical respiratory samples from known COVID-19 patients were collected from the Department of Microbiology of The Chinese University of Hong Kong and the Department of Health of Hong Kong SAR for SAVED validation. SARS-CoV-2 RNA of these clinical samples were prepared by QIAamp viral RNA minikit (Qiagen) following the manufacturer's protocol. Sixty-nine SARS-CoV-2 clinical samples with known lineages, $C_T$ value of RT-qPCR, and 10 NTC were included for platform validation (see Table S2 in the supplemental material). The information of these samples was kept blinded for staff conducting validation experiments.

Ethical approval of this study was given by the Joint Chinese University of Hong Kong–New Territories East Cluster Clinical Research Ethics Committee (The Joint CUHK-NTEC CREC) (CREC reference number 2020.076).

The preparation of crRNAs by IVT was modified from a previous protocol (25). The T7-3G IVT primer was annealed with the crRNA IVT templates by denaturation at 95°C for 5 min and slowly cooling down (0.1°C/s) to 4°C in a PCR thermocycler (Bio-Rad). The annealing reaction mixture contained 10 $\mu$M T7-3G IVT primer, 10 $\mu$M crRNA IVT templates, and 1× PCR buffer containing 10 mM Tris-HCl, 50 mM KCl, and 1.5 mM $MgCl_2$ (pH 8.3 at 25°C) for a total of 5 $\mu$L. The 5-$\mu$L annealing products were used as IVT templates by mixing with 5 $\mu$L rNTP buffer mix (5 mM each NTP final), 1 $\mu$L T7 RNA polymerase mix, and 9 $\mu$L water, incubated overnight at 37°C. The IVT products were treated by TURBO DNA-free kit and purified as above. The crRNAs were calculated by NEBioCalculator and diluted to 100 $\mu$M for storage.

**RPA and RT-RPA amplification.** TwistAmp liquid basic kit (TwistDx) was used for RPA and RT-RPA. The reaction mixture containing 10 $\mu$L 2× reaction buffer, 0.9 $\mu$L dNTPs (10 mM each; NEB), 2 $\mu$L 10× Basic E-mix, 0.96 $\mu$L forward and reverse RPA primers (10 $\mu$M), 1 $\mu$L RevertAid RT (200 U/$\mu$L; Thermo Fisher; used only for RT-RPA), 0.1 $\mu$L RNase H (10 U/$\mu$L; Thermo Fisher; only needed for RT-RPA), 1 $\mu$L 20× core reaction mix, 1 $\mu$L of 280 mM MgOAc, 2 $\mu$L RNA or DNA template, and water to 20 $\mu$L. The reaction mixture was incubated at 37°C for 30 min and heated at 75°C for 10 min. For the RPA and RT-RPA primer optimization, 1E+03 copies/$\mu$L of synthetic DNA/RNA were amplified. We compared the band or fluorescence signal to choose the best primer set, the selected RT-RPA, and additional primers shown in Table S4. For crRNA screening, a high concentration (1E+09 copies/$\mu$L DNA or RNA) of different samples was used as templates for RPA or RT-RPA to achieve better differentiation; thereafter, crRNA specificity was compared by CRISPR-Cas12a detection. For LoD determination, 1E+05 to 1E+0 copies/$\mu$L RNA of the corresponding variant and 1E+09 copies/$\mu$L RNA of wild type or different mutations at the same sites were used for RT-RPA and then detected by CRISPR-Cas12a using selected crRNAs.

**CRISPR-Cas12a detection.** The CRISPR-Cas12a detection mixture was modified from the previous protocols (19, 32, 42). Two different mixtures (low and high concentration) of Cas12a and crRNA were used for crRNA screening and CRISPR-Cas12a detection. The low concentration contained 1× NEB buffer 2.1, 40 nM Cas12a (NEB), 40 nM crRNA, and water to 17.6 $\mu$L (10% excess). The high concentration contained 1× NEB buffer 2.1, 100 nM Cas12a (NEB), 200 nM crRNA, and water to 17.6 $\mu$L (10% excess). Both mixtures were incubated at 37°C for 30 min; after incubation, 2.2 $\mu$L (10% excess) fluorophore-quenched ssDNA fluorescent reporter (1 $\mu$M, final concentration at 100 nM,/56-FAM/TTATTATT/3BHQ1/; Beijing Genomics Institute) and 2.2 $\mu$L (10% excess) of RPA/RT-RPA products were added. The 20-$\mu$L reaction mixtures were transferred to a 384-well plate (black, flat bottom; Corning) for fluorescent intensity monitoring. Fluorescent intensities were monitored every 5 min at 37°C for 1.5 h by Synergy H1 microplate reader (BioTeK) at FAM channel (excitation/bandwidth, 484/12.5; emission/bandwidth, 530/12.5). As for the fluorescent detection in tubes, the 0.2-mL 8-strip tubes were incubated at 37°C before visualizing by ChemiDoc touch imaging system (Bio-Rad) under UV light or blue LED light with a filter, and the detection was performed at 30 min and 1.5 h, respectively (32).

**Statistical analysis.** GraphPad Prism 9 (GraphPad Software, Inc.) was used for data analysis. One-way analysis of variance (ANOVA) test with Dunnett's multiple-comparison test was used for statistical analysis. Two-sided confidence intervals (95%) of sensitivity and specificity were analyzed by the Clopper-Pearson method. A $P$ value of <0.05 was considered significant.

## SUPPLEMENTAL MATERIAL

Supplemental material is available online only.
**SUPPLEMENTAL FILE 1**, PDF file, 3.4 MB.

## ACKNOWLEDGMENTS

This work was supported by internal grants from the Department of Microbiology, Faculty of Medicine, Chinese University of Hong Kong; Food and Health Bureau of the Hong Kong Special Administrative Region under Grant number COVID19F06 and COVID190107.

J.Y., N.B., and M.N.R. conceptualized this study under the supervision of M.I. T.F.T. assisted initial experiments for the development of CRISPR-Cas12a-based SARS-CoV-2 variant detection. J.Y., N.B., C.L., N.L., and K.Y.Y. performed the experiments and/or validated the platform on SARS-CoV-2 clinical samples. M.I., P.K.S.C., X.Y., Y.-Y.C., A.K.L.T., R.C.W.C., and E.C.-M.L. provided clinical samples, materials, and the analytical results of the samples. J.Y. made the initial draft of the manuscript with guidance from M.I., and all authors contributed to and agreed with the final version of the manuscript.

We declare that we have no conflict of interest.

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
