## [Reviewer comments · Microbiology Spectrum]

Microbiology Spectrum

Rapid SARS-CoV-2 Variants Enzymatic Detection (SAVED) by CRISPR-Cas12a

Jun Yang, Nilakshi Barua, Md Nannur Rahman, Carmen Li, Norman Wai-Sing Lo, Kai Yan Yeong, Tsz Fung Tsang, Xiao Yang, Yuk-Yam Cheung, Alan Tsang, Rickjason C. W. Chan, Eddie Leung, Paul Chan, and Margaret Ip

Corresponding Author(s): Margaret Ip, Chinese University of Hong Kong

Review Timeline:

Submission Date:	August 18, 2022
Editorial Decision:	September 18, 2022
Revision Received:	September 29, 2022
Accepted:	October 7, 2022

Editor: Maria Grazia Cusi

Reviewer(s): Disclosure of reviewer identity is with reference to reviewer comments included in decision letter(s). The following individuals involved in review of your submission have agreed to reveal their identity: Kaan Çeylan (Reviewer #2)

Transaction Report:

DOI: <https://doi.org/10.1128/spectrum.03260-22>

September 18, 2022

Prof. Margaret Ip
Chinese University of Hong Kong
Microbiology
Prince of Wales Hospital,
Ngan Shing Street
Shatin
Hong Kong

Re: Spectrum03260-22 (Rapid SARS-CoV-2 Variants Enzymatic Detection (SAVED) by CRISPR-Cas12a)

Dear Prof. Margaret Ip:

Link Not Available

Sincerely,

Maria Grazia Cusi

Journals Department
Reviewer comments:

Reviewer #1 (Comments for the Author):

Dear authors,

This manuscript (Ms#Spectrum03260-22) entitled "Rapid SARS-CoV-2 Variants Enzymatic Detection (SAVED) by CRISPR-Cas12a" by Yang et al. describes Cas-12a can be used for the rapid

differentiation of SARS-CoV-2 variants. Indeed, the specificity and accuracy of the present assay using blind clinical samples would be applicable to clinical settings. I'll provide you with a few comments below.

Comment-1 (Table 1): Too busy to compare the difference between mutation sites. Please modify Table 1 in a reader-friendly style.

Comment-2 (Figures 1o, 1u, S15, S21): As the authors mentioned in Legend for Figure1, the assay for T478K and N501Y was conducted using RNA templates because the use of DNA templates resulted in non-specific amplification. Why did this inappropriate reaction occur?

Comment-3: Please describe the reason why you set the fluorescence threshold as 5000 AU.

Comment-4 (Figure 4): Some mutation sites such as T478K, E484A, and L452R showed no dose-dependency. The authors should discuss why the present assay resulted in the tendency.

Reviewer #2 (Comments for the Author):

Dear author,

Your work titled Rapid SARS-CoV-2 Variants Enzymatic Detection (SAVED) by CRISPR-Cas12a has been reviewed by me. Congratulations for the hard work you put into it. While the text is generally acceptable, some points may be challenging for the reader.

First of all, since the tables are very long, it will be difficult for the reader to follow.

It becomes difficult to understand the subject, as repetitive expressions are included in the findings and discussion part.

In addition, since two different molecular techniques are compared, it would be appropriate to mention the cost difference between them in the study.

I would appreciate it if you would consider these issues.

Good work.

Staff Comments:

Preparing Revision Guidelines

Please return the manuscript within 60 days; if you cannot complete the modification within this time period, please contact me. If you do not wish to modify the manuscript and prefer to submit it to another journal, please notify me of your decision immediately so that the manuscript may be formally withdrawn from consideration by Microbiology Spectrum.

If your manuscript is accepted for publication, you will be contacted separately about payment when the proofs are issued; please follow the instructions in that e-mail. Arrangements for payment must be made before your article is published. For a

complete list of **Publication Fees**, including supplemental material costs, please visit our website.

Dear Reviewers,

Thank you very much for your kind comments. Your suggestions are highly appreciated, and we have modified the manuscript according to your comments. Please check the following point-by-point response.

Link Not Available

Response: The files have been prepared according to your requirements.

Response: All data has been verified.

The ASM Journals program strives for constant improvement in our submission and publication process. Please tell us how we can improve your experience by taking this quick Author Survey.

Response: None.

Sincerely,

Maria Grazia Cusi

Journals Department
Reviewer comments:

Reviewer #1 (Comments for the Author):

Dear authors,

This manuscript (Ms#Spectrum03260-22) entitled "Rapid SARS-CoV-2 Variants Enzymatic Detection (SAVED) by CRISPR-Cas12a" by Yang et al. describes Cas-12a can be used for the rapid differentiation of SARS-CoV-2 variants. Indeed, the specificity and accuracy of the present assay using blind clinical samples would be applicable to clinical settings. I'll provide you with a few comments below.

Comment-1 (Table 1): Too busy to compare the difference between mutation sites. Please modify Table 1 in a reader-friendly style.

Response: Many thanks for your comments. We have replaced the original Table 1 with the original Figure S1. Kindly check Figure 1.

Comment-2 (Figures 1o, 1u, S15, S21): As the authors mentioned in Legend for Figure1, the assay for T478K and N501Y was conducted using RNA templates because the use of DNA templates resulted in non-specific amplification. Why did this inappropriate reaction occur?

Response: Many thanks for your comments. The primers of T478K and N501Y have mismatch(es) for the Omicron variant containing S477N+T478K and Q498R+N501Y. RT-RPA by T478K and N501Y primers using RNA samples decreased amplification

compared RPA using DNA samples. The fluorescence signal between T478K and S477N+T478K, N501Y and Q498R+N501Y already showed obvious difference by RPA following CRISPR-Cas12a detection, with the help of reverse transcription, the signal of S477N+T478K and Q498R+N501Y lower than the threshold, which enables these mutation sites differentiation. Kindly check lines 508-512 of the marked-up manuscript or lines 399-403 of the revised manuscript.

Comment-3: Please describe the reason why you set the fluorescence threshold as 5000 AU.

Response: We used 5000 AU as the threshold and validated against spectrophotometry because below 5000 AU the signal was not visible under the UV light. Kindly check lines 282-283 of the marked-up manuscript or lines 174-175 of the revised manuscript.

Comment-4 (Figure 4): Some mutation sites such as T478K, E484A, and L452R showed no dose-dependency. The authors should discuss why the present assay resulted in the tendency.

Response: Many thanks for your critical comment. Though there were some fluctuations on the fluorescence value compared to the Ct value, our platform's clinical samples were positive. As long as it was positive, there was an agreement between our platform and RT-qPCR. The fluctuations has been also observed in the experiments conducted by de Puig et al., 2021; Patchsung et al., 2020; Broughton et al., 2020; Liang et al., 2021 (The reference details and figures are mentioned in the following table for your kind reference).

Figure No.	References
Figure 4	de Puig H, Lee RA, Najjar. Minimally instrumented SHERLOCK (miSHERLOCK) for CRISPR-based point-of-care diagnosis of SARS-CoV-2 and emerging variants. Sci Adv , 2021 ;7. DOI: 10.1126/sciadv.abh2944
Figure 4	Patchsung M, Jantarug K, Pattama A, et al. Clinical validation of a Cas13-based assay for the detection of SARS-CoV-2 RNA. Nat Biomed Eng , 2020 ;4:1140–1149. DOI: 10.1038/s41551-020-00603-x.
Figure 2e	Broughton JP, Deng X, Yu G, et al. CRISPR–Cas12-based detection of SARS-CoV-2. Nat Biotechnol , 2020 ;38:870–

	874. DOI: 10.1038/s41587-020-0513-4
Figure 5	Liang Y, Lin H, Zou L, et al. CRISPR-Cas12a-Based Detection for the Major SARS-CoV-2 Variants of Concern. Microbiol Spectr , 2021 ;9:1–19. DOI: 10.1128/Spectrum.01017-21

Reviewer #2 (Comments for the Author):

Dear author,

Your work titled Rapid SARS-CoV-2 Variants Enzymatic Detection (SAVED) by CRISPR-Cas12a has been reviewed by me. Congratulations for the hard work you put into it. While the text is generally acceptable, some points may be challenging for the reader.

First of all, since the tables are very long, it will be difficult for the reader to follow.

Response: Thank you for your comments. We have replaced the Table 1 with Figure 1, which is much convenient for readers to follow. For the original Table 2 and Table 3, now they are Table 1 and Table 2 in the marked-up manuscript. As so many mutation sites need to be mentioned, it will lose important information if shortened.

It becomes difficult to understand the subject, as repetitive expressions are included in the findings and discussion part.

Response: We have deleted the respective repetitive expressions in following lines:

Lines 347-348: “The result of 30 min incubation was the same as 1.5 h, thus, here only shows 30 min incubation.”

Lines 379-382: “we used 69 clinical samples and 10 no template controls (NTC); The samples included Wild type (n=5), Alpha (n=10), Beta (n=10), Delta (n=20), Omicron BA.1 (n=5), and BA.2 (n=19) variants of SARS-CoV-2.”

Lines 387-389: “The sequences, variant types, and Ct value of clinical samples were unknown to the researcher until the completion of analyzing SAVED CRISPR-Cas12a-based detection data to minimize interpretation bias.”

Lines 453-457: “We opted for the four strategies to optimize crRNAs for detection. In this study, all selected crRNAs targeting SNP belong to chimeric or short crRNAs strategies (Figs. 1-2 and Table 1). Mutation sites with deletions or insertions can be differentiated by regular length spacer, otherwise, at least 18-nt can be used as many mismatches were present (Fig. 1d,e,f,h,i).”

Lines 463-465: “As we reported previously, when we used 24-nt spacer, 20-nt spacer, and chimeric crRNAs, the specificity of chimeric crRNA was the strongest, whereas the signal was the weakest (Figs. S11-17,21) [32].”

In addition, since two different molecular techniques are compared, it would be appropriate to mention the cost difference between them in the study.

Response: We added the cost difference between our platform and RT-qPCR in Table S3 and lines 429-430 of the marked-up manuscript or lines 320-321 of the revised manuscript.

I would appreciate it if you would consider these issues.
Good work.

Staff Comments:

Preparing Revision Guidelines

Response: The files have been prepared according to your requirements.

For complete guidelines on revision requirements, please see the journal Submission and Review Process requirements

at <https://journals.asm.org/journal/Spectrum/submission-review-process>. Submissions of a paper that does not conform to Microbiology Spectrum guidelines will delay acceptance of your manuscript. "

Response: The files have been prepared according to your requirements.

Please return the manuscript within 60 days; if you cannot complete the modification within this time period, please contact me. If you do not wish to modify the manuscript and prefer to submit it to another journal, please notify me of your decision immediately so that the manuscript may be formally withdrawn from consideration by Microbiology Spectrum.

If your manuscript is accepted for publication, you will be contacted separately about payment when the proofs are issued; please follow the instructions in that e-mail. Arrangements for payment must be made before your article is published. For a complete list of Publication Fees, including supplemental material costs, please visit our website.

Corresponding authors may join or renew ASM membership to obtain discounts on publication fees. Need to upgrade your membership level? Please contact Customer Service at Service@asmusa.org.

October 7, 2022

Prof. Margaret Ip
Chinese University of Hong Kong
Microbiology
Prince of Wales Hospital,
Ngan Shing Street
Shatin
Hong Kong

Re: Spectrum03260-22R1 (Rapid SARS-CoV-2 Variants Enzymatic Detection (SAVED) by CRISPR-Cas12a)

Dear Prof. Margaret Ip:

Your manuscript has been accepted, and I am forwarding it to the ASM Journals Department for publication. You will be notified when your proofs are ready to be viewed.

Sincerely,

Maria Grazia Cusi
Editor, Microbiology Spectrum

Journals Department
Supplemental file 1: Accept